# Harmful Microalgae Detection: Biosensors versus Some Conventional Methods

**DOI:** 10.3390/s22093144

**Published:** 2022-04-20

**Authors:** Jeremy Jason Chin Chwan Chuong, Mahbubur Rahman, Nadiah Ibrahim, Lee Yook Heng, Ling Ling Tan, Asmat Ahmad

**Affiliations:** 1Southeast Asia Disaster Prevention Research Initiative (SEADPRI), Institute for Environment and Development (LESTARI), Universiti Kebangsaan Malaysia, Bangi 43600, Selangor Darul Ehsan, Malaysia; p118359@siswa.ukm.edu.my (J.J.C.C.C.); p110052@siswa.ukm.edu.my (N.I.); lingling@ukm.edu.my (L.L.T.); 2Department of Chemical Sciences, Faculty of Science and Technology, Universiti Kebangsaan Malaysia, Bangi 43600, Selangor Darul Ehsan, Malaysia; mahbubur.ged@diu.edu.bd; 3Department of General Educational Development (GED), Faculty of Science & Information Technology, Daffodil International University, Dhaka 1341, Bangladesh; 4School of Biosciences and Biotechnology, Faculty of Science and Technology, Universiti Kebangsaan Malaysia, Bangi 43600, Selangor Darul Ehsan, Malaysia; drasmart@gmail.com

**Keywords:** harmful algae, red tide, biosensor, HAB detection method, nanomaterial, conventional method

## Abstract

In the last decade, there has been a steady stream of information on the methods and techniques available for detecting harmful algae species. The conventional approaches to identify harmful algal bloom (HAB), such as microscopy and molecular biological methods are mainly laboratory-based and require long assay times, skilled manpower, and pre-enrichment of samples involving various pre-experimental preparations. As an alternative, biosensors with a simple and rapid detection strategy could be an improvement over conventional methods for the detection of toxic algae species. Moreover, recent biosensors that involve the use of nanomaterials to detect HAB are showing further enhanced detection limits with a broader linear range. The improvement is attributed to nanomaterials’ high surface area to volume ratio, excellent biological compatibility with biomolecules, and being capable of amplifying the electrochemical signal. Hence, this review presents the potential usage of biosensors over conventional methods to detect HABs. The methods reported for the detection of harmful algae species, ranging from conventional detection methods to current biosensor approaches will be discussed, along with their respective advantages and drawbacks to indicate the future prospects of biosensor technology for HAB event management.

## 1. Introduction

The ocean and freshwater constitute a global source of life, containing a diverse range of marine and freshwater phytoplankton species, such as algae and cyanobacteria. These organisms serve an important role in the aquatic biological environment by supplying oxygen and food for other species. However, algal bloom events can be triggered due to adequate nutrients discharged from agriculture, domestic, and industry [1,2,3,4,5]. Furthermore, the combined parameters of temperature, coastal development, water column structure, and eutrophication are also likely to cause algal blooms [6,7]. The increasing incidence of algae blooms is fast becoming a global concern as some algae species can produce toxins that could endanger aquatic life and human [8,9]. Hence, the proliferation of microalgae that produces harmful toxins is known as harmful algal blooms (HABs).

The intrinsic factor of freshwater and marine ecosystems are the results of photosynthetic biomass, such as phytoplankton and zooplankton, being in significant quantity. This biomass is regarded as food for a huge population of aquatic inhabitants [10,11]. Among the different types of photosynthetic biomasses, microalgae are one of the major producers due to their high photosynthetic activities and their place in the food chain of aquatic organisms [8,11]. Additionally, among the 5000 species of aquatic algae that have been discovered, over 300 species are known to proliferate or bloom rapidly and subsequently affects the colour of the sea surface [2,8].

HAB poses severe threats to marine ecosystems and humans and its effect is intensifying as a result of increased seafood consumption, growth in the number of coastal inhabitants, and tourism [12,13,14,15]. HABs can produce hazardous toxins, mainly neurotoxins that can be introduced into the human body by the consumption of contaminated fish, seafood products, or water [16,17]. These activities are highly contributing to human exposure to HAB which can affect digestive tract problems, respiratory illness, memory loss, seizures, lesions, and skin irritation [18,19,20]. In some cases, high exposure to HAB can cause mortality. The estimated health cost for mild, moderate, and severe digestive illnesses was found to be USD 86, USD 1015, and USD 12,605, respectively [21]. Whereas treatment for respiratory illness ranging from mild, moderate, and severe cases will cost about USD 86, USD 1235, and USD 14,600, respectively.

Due to the possible impact of HAB, it is vital to have HAB monitoring techniques that are rapid and accurate in identifying and assessing aquatic pathogens, environmental phenomena, seaside dynamics, and processes that can influence ocean ecosystems [22,23,24]. A broad range of methods and techniques that were detailed in the last decade are largely based on the conventional methods of HAB detection. However, conventional methods generally hold several disadvantages, such as being costly, time-consuming, lengthy experiments, and so on.

The detection technology for harmful microalgae is currently mostly using morphological characteristics of algal cells, traditional microscopic examination technique (MET) and various techniques based on high-performance liquid chromatography (HPLC), absorption spectral analysis (ASA), and fluorescence spectral analysis (FSA). As pointed out by Liu et al. [25] in a review published recently, these methods have their strengths and weaknesses. With respect to their weaknesses, for example, HPLC analysis and molecular detection methods, such as fluorescence in situ hybridization (FISH), sandwich hybridization assay (SHA), and quantitative polymerase chain reaction (qPCR) require sophisticated instruments and professional operation, thus not suited for on-site rapid detection. Methods based on ASA and FSA have a poor resolution, which means that they could only differentiate between algal species at the phylum level. Immunological technology, such as immunofluorescence assay (IFA), immunosensing assay (ISA), and enzyme-linked immunosorbent assay (ELISA) have been developed for the detection of harmful microalgae but they have the weakness of troublesome in the preparation of antibodies.

Although the needs for the detection of harmful microalgae species vary largely, the main desirable methods for the detection are the accuracy, reliability, and convenience for operation. Equally important is the portability and the implementation of intelligence and digitization for newly emerging technologies. Thus, rapid, high efficiency, simplicity, and automation will be the ultimate objective of a detection technology for harmful microalgae. All this will contribute to the real-time and rapid detection of harmful microalgae in the field. This is likely to be fulfilled by biosensor technology that is currently developing rapidly. Biosensors represent an attractive alternative to environmental monitoring that allows us to circumvent the drawbacks of conventional methods. Biosensors generally show high specificity, wide linearity, low detection limit, portability, and comparatively low cost. Nevertheless, there have been very few advancements in biosensors for the detection of toxic algae species. Additionally, limited reports can be found on detecting harmful algae with biosensors fabricated with nanomaterials that can further improve the detection limit and linear range capacity. 

In this review, we focus on discussing the current development of biosensor technology for harmful microalgae detection. In addition, the importance of the biosensor methods is compared with several conventional methods for detecting harmful microalgae, especially those involved in alga bloom environment (HAB). The benefits and drawbacks of each method will be explored to find a better approach for HABs detection and monitoring. The characteristics of classifying the advantages and drawbacks will mainly be based on detection limits, sensitivity, accuracy, and simplicity of each respective method. Figure 1 shows the general overview of the purpose of this review.

## 2. Conventional Methods for HAB Identification

The main conventional methods in HAB monitoring and detection is based on light microscopy and counting chamber [26]. Light microscopy can be utilized to do species identification among the various types of HAB, but is not accurate. Whereas counting chamber will be used to estimate the quantity of HAB species found in samples collected. Other approaches in HAB identification can be based on mouse bioassay (MBA) or chromatographic technique. Although, these two mentioned approaches focus more on detecting the HAB toxins content and is thereby unsuitable for early warning system of HAB event but only to monitor the toxicity of the water. Alternatively, many species from HAB, in particular, and phytoplankton in general, can be detected via molecular methods, which are fast and accurate methods capable of simultaneous qualitative detection [27]. Molecular methods can also identify and quantify harmful algae species [28,29]. The biomedical research and diagnostic industries have been designing simplified ways to sample preparation and distribution while utilizing molecular probe technologies to interpret results [30]. For instance, sample preparation and analysis systems that are portable for in situ detection have been designed but no major implementation had occurred [31]. Although other techniques used for the same purpose include DNA microarrays with different molecular probe techniques [32,33], when it comes to identifying phytoplankton, molecular approaches are considered to be quicker and more precise than light microscopy [34]. Several types of molecular methods, including fluorescent in situ hybridization (FISH) of whole-cell, polymerase chain reaction (PCR)-based assays, sandwich hybridization assays, enzyme-linked immunosorbent assay (ELISA) colorimetric whole-cell analysis, monoclonal antibody probes, and DNA microarrays are capable of identifying algae species and detecting toxic algae in routine monitoring programs [24,25,28,35].

### 2.1. Microscopic Analysis

Globally, nations have employed HAB monitoring programs as warning systems. In the last decade, one of the main methods of monitoring and mitigation depends on visually inspecting water discolouration, dead fishes, and laborious cell counts [34]. Samples will be required to be dispatched to a laboratory for testing purposes to examine for the presence of harmful compounds using tried-and-tested procedures, such as light microscopy (LM) [13]. However, heterotrophs and autotrophs with sizes below 5 μm will be extremely difficult to be distinguished under LM [36]. For instance, *Pseudo-nitzschia* species cannot be identified at the species level via LM [11], whereas the high difficulty is present in species identification among *Alexandrium* variations. Furthermore, the counting of HAB cells is completed by humans, and this will inevitably result in human error mistakes that could lead to inaccurate results being produced. Thereby, the drawback of LM would be its incapability to discern minor differences between species. Both transmission electron microscopy (TEM) and scanning electron microscopy (SEM) are capable of doing identification of various pico- and nano-sized organisms but will inevitably increase overall expenses and time [27]. Whereas epifluorescence microscopy (EFM) does allow the sizing distinction among heterotrophs and autotrophs. EFM also provides a more accurate enumeration than LM but still holds difficulty in species identification [36].

Optical microscopy with normal light or epifluorescence would be used to identify paralytic shellfish poison (PSP) via *Alexandrium* (dinoflagellate) and other HABs in marine waters [37]. This approach is highly successful but is restricted by the sheer amount of samples that could take several days to complete analysing high numbers of samples. The examination of harmful algae species from water samples will require a rough estimation of 2 h on average. This will be reflected as an estimation work rate of processing 20 samples per week by one person [11]. Sometimes filtration of water samples is necessary during the sample preparation phase with 4 μm pore size filters and air dry the cells in immersion oil for 2–4 days before being examined under standard LM [36]. These time lags will impede the early warning systems of bloom events, as numerous samples are required from the different sites of the water body to accurately conclude if there will be a possibility of HAB. A simplified diagram is illustrated in Figure 2 to show the flow of microscopy approaches in HAB examination.

An alternative method to quantifying HAB content will be based on identifying the chlorophyll concentration in the sample [38]. The quantity can only be estimated if the biomass to chlorophyll ratio is known beforehand. In addition, flow cytometry can be utilized in calculating HAB cell concentrations but the downside is the fact that organism’s sizing is not precise [36]. This approach is only useful for homogenous populations that have distinctive pigment or size and is not successful in organisms with a wide range of sizes and shapes in field samples [39]. Despite all the disadvantages stated, the microscopical method does hold an advantage in allowing excellent biological detail in the whole size spectrum of HAB species and other microbial autotrophs [36]. Although, due to the mentioned drawbacks, improvement in monitoring of HAB for on-site and near-real-time water analyses are desired.

### 2.2. Fluorescent In Situ Hybridization (FISH)

This approach relies on the principle of oligonucleotide such as DNA or PNA as the probe that is attached with a fluorescent marker and required to penetrate through cells to hybridize with the target sequence. The targeted cells will fluoresce and can be visually detected with epifluorescence microscopy (Figure 3). Studies can be found on applying FISH to detecting certain HAB species, but several disadvantages are present. For instance, before observing fluorescently labelled cells, several necessary purification steps including: (I) sample treatment, (II) centrifugation, (III) pipetting, and (IV) washing may lead to the loss of target cells [40]. This will lead to unreliable quantification results unless a gentle filtration method is applied. In addition, the loss of cells from purification steps, the duration from sample collection to reaching a laboratory may gradually decompose the rRNA in the cells [41]. This can reduce the fluorescent intensity of labelled cells. Furthermore, different types of cell walls and membranes can be found in marine phytoplankton and this causes the difficulty in having a universal FISH protocol in fixing all sorts of microalgae cells [42]. For instance, certain modified saline ethanol can allow high permeability for a probe to enter and access target sequence in some microalgal species but does not work well with *Alexandrium* species, where auto-fluorescence was observed [43]. The background noises can mask over target fluorescent and make results hard to be interpreted.

The FISH approach was investigated to detect *A. minutum* harmful algae species with ribosomal DNA probes [43]. Their results showed high specificity towards *A. minutum* and no cross-reactivity with other *Alexandrium* species was observed. However, their methodology involved multiple centrifugation and washing steps that may cause some loss of microalgae cells. Whereas another developed FISH that used a specific PNA probe can detect *Prorocentrum donghaiense* [44]. Their findings found that the PNA probe is more sensitive compared to the DNA probe in detecting the harmful algae species as fluorescent is more intensive with the PNA probe. Although, the quantification of harmful algae cells is only as liable to that of the LM approach and is rather time-consuming.

### 2.3. Polymerase Chain Reaction (PCR)-Based Assays

PCR based techniques generally involve the amplification of targeted sequence for several cycles and are verified for the presence of targets. If the target sequence is present, the polymerase will duplicate the sequence defined by specific pair of primers that had hybridized with the sample DNA (Figure 4). After several cycles of amplification, the sequence concentration will increase and become easier to be detected by any sort of potential method. A multiplex PCR (mPCR) developed for the multiple detections of six common HAB species found in the coastal region of China could detect as low as 0.06 ng/μL of target DNA [45]. Based on the mPCR system, the detection limit for *Karlodinium veneficum* (Kv), *Prorocentrum donghaiense* (Pd), and *Karenia mikomotoi* (Km) was 60 cells whereas *Chattonella marina* (Cm), *Skeletonema* spp., and *Scrippsiella trochoidea* (St) were 6 cells. However, this was based on pre-set sample concentration as the developed mPCR is a non-quantitative assay and could not provide accurate cell density reading. This mPCR approach is however useful in simultaneous detection of the presence of multiple HAB species and could be used for recognition purposes. The qualitative detection is rather sensitive but numerous optimization steps are required to be implemented. This includes primer design, primer concentration, Taq DNA polymerase concentration, dNTP concentration, Mg^2+^ concentration, and annealing temperature. For instance, when designing primer, the formation of primer dimer should be avoided and the annealing temperature should be suitable for all six types of primers. Studies will also need to be carried out to select the ideal primer with low mutual interference. Determining the Mg^2+^ concentration is also necessary for the efficient mPCR system to function well.

Quantitative PCR or simply qPCR is a technique that amplifies and detect a specific DNA sequence. This is, thereby, an improvement over conventional PCR which requires other methods to be implemented for quantification purposes. The inclusion of fluorescent probes or dye is crucial for the monitoring of amplified sequence via fluorimeter and generating real-time qPCR curve. The linear exponential phase will be used for quantification and comparing with standard curve allows possible calculation of the number of target DNA that was present in the initial sample [46]. The type of fluorescent, such as SYBR Green, is relatively inexpensive, which is an advantage to reduce cost, but holds a major drawback in possible binding to all types of double-stranded DNA, such as primer-dimer and makes quantification to be less reliable [47]. Whereas the TaqMan fluorescence probe has superior specificity and makes quantification to be more reliable, however, the cost is more expensive [48]. A qPCR that utilized a TaqMan hydrolysis probe as the detection chemistry could quantify *Alexandrium* spp. with a detection limit of 5 cells L^−1^ [49]. Despite the use of superior fluorescence probes, qPCR assays, in general, may provide overestimates of abundance. This can be reasoned with the possible decreasing recovery of DNA as cells number used for DNA extraction became lower [49]. qPCR does have an advantage in being rapid and high-throughput as many qPCR machines can operate on 96 or 384 well plates. However, in terms of enumeration, qPCR generally have higher sensitivity compared to a haemocytometer and counting chamber approaches [46]. Although, the major drawback of qPCR will be its incapability to have a generic assay to detect all the toxin-producing HAB species while excluding the non-toxic species [46]. 

### 2.4. Enzyme-Linked Immunosorbent Assay (ELISA)

An ELISA technique commonly involves the use of an enzyme that will catalyse substrates that are initially bound toward targets to cause a fluorescent signal (Figure 5). This technique can also be seen in detecting and quantifying HAB species. A study that used ELISA to detect the rRNA of harmful dinoflagellate *A. minutum* was conducted and a sandwich hybridization assay (capture probe and reporter probe) was used in this method [23]. The results were analysed based on colorimetric detection in the presence of an anti-digoxigenin antibody conjugated with horseradish peroxidase that can react with a specific substrate to produce a green colorimetric product. The advantage of this molecular biological approach can detect the *A. minutum* cells at varying cell counts even in the presence of complex background. However, the accuracy of quantifying the cell numbers is only based on estimation due to the limited correlation of signal product to cell numbers. Additionally, *A. minutum* species could also be detected using monoclonal antibody methods and colorimetric whole-cell ELISA [50]. This technique demonstrated a comparatively low detection limit, high sensitivity, and specificity for the analysis of natural seawater samples. Although, the drawbacks of this method include the use of rats to generate monoclonal antibodies that are complementary to the antigen of *A. minutum* and the results produced a lack of accuracy in quantifying the cell density.

### 2.5. Microarray

Microarrays typically contain specific probes that are applied on their surface in order to potentially detect thousands of targets in a single test (Figure 6). This makes microarrays to be a powerful molecular tool for detection purposes. Fluorescently labelled target sequences that bind to its complementary probe will be identified by a laser that excites the fluorescent dye and the emission level is detected by a detector. Steps involved in a microarray experiment include: (I) production of microarray, (II) isolation and preparation of nucleic acid, (III) hybridization, and (IV) data analysis [42]. A bead-based fibre optic microarray that contains oligonucleotide probes specific for rRNA of target HAB species could detect as low as 5–10 cells per sample for *Pseudo-nitzchia australis* and *A. fundyense* while showing a detection limit of 50 cells per sample for *A. ostenfeldii* [51]. Their experiment proved the usefulness of simultaneous detection of several HAB species and is a major advantage compared to other common conventional methods which typically involve one method for one species identification. Another fibre optic microarray which targets the same three species of HAB had shown a limit of detection of 5 cells per sample and the detection time is within 45 min, which is considerably fast [52].

The simplicity to apply a sample to the sensor array while being capable of being reused without significant loss of sensitivity is another attractive feature of microarray [51]. The dehybridization process can be simply achieved by washing with formamide and may proceed with the next sample experiment. Less than 2% degradation was observed with over 200 hybridization and dehybridization cycles for fibre arrays [53]. The complication with this methodology will be to interpret the results computerized by a charged-couple device camera that detect the exciting light from fluorescent. Proper interpretation of the results is necessary to accurately quantify the HAB cell density in the sample used. Moreover, microarrays are stated to have higher experimental and set-up costs as compared to qPCR [46]. This could be reasoned with the large number of probes needed to detect and differentiate different species of organisms. Additionally, experts are needed to design high specificity probes towards target organisms while also considering the possibility of cross-reactivity among different probes in a multiplex. Table 1 summarized several of the reported molecular biological methods in HAB detection.

## 3. Biosensor Methods

Biosensor methods involve a biochemical recognition component that is combined with a signal transducer to detect specific targets. The recognition or bio-receptor component, such as specific probe sequence, antibodies, or enzymes could specifically bind to the target of interest or catalyse a biochemical reaction [54]. The bio-recognition event will then be converted into a quantifiable signal via a transducer (Figure 7). Biosensors are attractive candidates to overcome the limitations of traditional detection quantification methods due to their accuracy, simplicity of use, user-friendly, cost-effective, robustness, low power requirements, as well as their rapid turnaround time, high sensitivity, and versatility [55]. In addition, biosensor has the capacity for miniaturization and in-field application, e.g., portable device, which is suitable to improve monitoring methods by allowing for rapid on-site identification of microbiological pollutants. Biosensor design that could perform multiple targets detection concurrently can be an advantage for reducing the sample required [56].

Biosensors can be categorized into piezoelectric biosensor, calorimetric biosensor, electrochemical biosensor, and optical detection, which are all according to different types of transducer. Electrochemical and optical biosensors are the most widely used among other types of biosensors due to the ease of operation, high sensitivity, and enable direct, real-time, and label-free detection [57,58]. The application for electrochemical and optical detection of HAB has been further discussed in the section below.

### 3.1. Electrochemical Biosensor Methods

The fundamental principle of the electrochemical biosensor is the chemical interactions between immobilized biocomponents and analyte that produces or consume ions or electrons, changing the measurable electrical properties of the solution, such as potential or electric current (Figure 8) [59]. Electrochemical biosensors with increased specificity, stability and sensitivity that are small and easy to fabricate are now accessible [60]. Furthermore, electrochemical biosensors are highly sensitive, portable, relatively inexpensive, and simple to build [61]. Therefore, the electrochemical DNA biosensor can serve as an appropriate HAB monitoring program.

Two detection methods, i.e., ‘Rapid PCR-Detection’ and ‘Hybrid PCR-Detection’ were introduced for targeted detection (Table 2). The ‘Rapid PCR-Detection’ identifies and quantifies PCR products through biotin and fluorescein labelling during PCR without a hybridization step. ‘Hybrid PCR-Detection’, on the other hand, involves the increased specificity via hybridization to a DNA probe. However, this method based on a single-plex reaction was only capable of identifying single-base mutations in nucleic acids isolated from pure culture [62]. An 8-plex assay was designed based on the two detection methods for multi-target detection of microbial contaminations and also dinoflagellate *Karenia brevis (K. brevis)* in coastal waters [63]. Whereas a handheld DNA biosensor approach, based on sandwich hybridization and molecular DNA probes, was capable of detecting the ribosomal RNA (rRNA) of harmful algae *A. ostenfeldii* [32,64]. This portable biosensor simplifies the detection of harmful algae, however manual RNA separation and manipulation of the hybridization procedures are required [63,64].

Another monitoring system, a multi-probe chip, and a semi-automated rRNA biosensor were employed in conjunction with a sandwich hybridization assay approach for in situ detection of harmful algae [23]. The target organism is identified by specific capture and signal probes to the harmful algae rRNA. The capture probes are immobilized on the surface of the sensor chip and the signal probe is coupled to digoxigenin, which binds to an antibody-enzyme complex. The enzyme catalyses a redox reaction that can be measured as an electrochemical signal using a potentiostat. The main drawback of this procedure was the isolation of rRNA from the target organism, and most preparatory stages were completed manually in a well-equipped laboratory [23,64]. To address the mentioned limitation, a semi-automated device, known as ALGADEC, was developed for in situ analysis of potentially harmful algae [36]. The semi-automated part highlights the use of lysis protocol instead of manual RNA separation. The detection approach involves a sandwich hybridization with capture probe and signal probe (digoxigenin-labelled), and the resulting electrical current can be measured upon substrate addition.

Furthermore, an electrochemical DNA biosensor based on a double DIG-enzymatic label and direct HRP-labelled signal probe could detect three species of HABs [65]. The electrochemical signal was examined using cyclic voltammetry or by simply checking the amperometric current magnitude. This amperometric procedure was then investigated in detecting HABs with mixed self-assembled monolayer and bovine serum albumin as a blocking agent in the electrochemical signal [34]. The biosensor’s performance was improved in terms of greater sensitivity and enhanced detection limit.

Molecular recognition in DNA biosensors is accomplished by hybridization of the target sequence with the complementary probe. The ability of a biosensor to directly recognize nucleic acids in complex samples is a significant advantage over other techniques, such as polymerase chain reaction (PCR) or ELISA, which needs extensive purification and amplification procedures [34]. The detection methods are usually carried out by using electrochemical and optical biosensors [67,68]. Both the electrochemical and optical biosensors approaches can directly identify nucleic acids from complex samples without the need for target purification and amplification [69].

Previously established DNA biosensors for the detection of *K. brevis* had achieved an accurate detection limit of 1000 cells L^−1^ within 3 h to 5 h [63], while another DNA-based sensor was capable of detecting 58 µg of synthetic *Prymnesium parvum* [34]. The biosensors used for both studies were based on electrochemical transducers.

Immunosensors are analytical devices that detect antigen–antibody interaction via coupling of immunochemical response to the surface of a device. There are two important factors in this context, which include the immobilization of antibodies on the sensor surface and the efficient generation of electrochemical signals [70]. The former involves antibody orientation vis-a-vis antigen binding sites. The literature points out the construction of electrochemical immunosensors based on carbon electrodes. In recent years, immunosensors with precise detection of target antigen have sparked attention as a tool for environmental evaluations, clinical diagnostics, food control, and industrial monitoring [71]. Additionally, immunosensors generally possess high sensitivity and specificity in the diagnosis field [54] and are thereby suitable for HAB detection.

Electrochemical immunosensors are capable of detecting analytes in complex biological media with high specificity and sensitivity. For instance, a designed electrochemical immunosensor was capable of detecting as low as 1 cell of *A. minutum* harmful algae in a millilitre of water sample with minimum cross reaction with non-toxic algae species [72].

In terms of detection time, the older model of biosensor designs [62,64] had a rather slow response time that require more than 4 h, while newer biosensors [63,73] have improved detection time and showed good response time of within 2 h [23,35,63,65]. Thereby, optimization and implementation of suitable biosensor design can possibly further improve the detection time.

### 3.2. Optical Biosensors

Optical chemical sensors have also been employed to detect HAB [74]. Optical chemical sensors are based on the reaction between biomolecule compounds and analytes, where the optical characteristics are induced by UV-vis absorption, bioluminescence, chemiluminescene, fluorescence, or reflectance. These biosensors have the benefits of being flexible, compact, and resistant to electrical noise (Figure 9) [75]. Furthermore, thanks to the use of optical fibre, it is feasible to construct very compact sensors, making them suited for measurement [73,76].

Unlike toxins from microalgae, to date there are not many optical biosensors for the determination of microalgae species. So far, *Alexandrium* species had been identified via an optical technique known as SPIRIT+, which utilizes portable surface plasmon resonance (SPR) biosensing equipment and peptide nucleic acid (PNA) probes [66]. SPR is a label-free technology that measures changes in the refractive index when a target sequence is hybridized on a surface. The resonance unit (RU) was used to quantify changes in the refractive index, where one RU is equal to 1 × 10^−6^ degrees. Additionally, SPIRIT is claimed to be a cost-effective and user-friendly approach that yields quick results.

## 4. Nanomaterials in Biosensor

Nanoparticles are defined as extremely small materials with diameters ranging from 1 to 100 nm. They offer unique features compared to bulk-sized materials and are, hence, widely used in biomedical, pharmaceutical, cosmetic, environmental, electronic, and energy fields [77,78]. Nanomaterials are generally divided into three categories, which include (i) zero-dimensional materials or also known as quantum dots, (ii) one-dimensional material, and (iii) two-dimensional materials. Different quantum dots typically vary in shape and diameter, while one-dimensional materials include nanorods and nanowires. As for two-dimensional material, it can include nanobelts, nanosheets, nanodiscs, and films.

The incorporation of nanoparticles into biosensors offers advantages in terms of fast and high-throughput detection. This can be reasoned by the great physical confinement of electrons at the nanoscale, which gives rise to high surface to volume ratio properties. As a result, nanoparticles are thought to be promising sensing materials.

### 4.1. Nanomaterial-Based Immunosensor

An electrochemical immunosensor that incorporated nanomaterials to detect *A. minutum* was successfully designed [75]. This biosensor was modified with gold nanoparticles (AuNPs) [79] and functionalized with specific nanobodies [80] for high sensitivity detection. The detection platform used was a glassy carbon electrode modified with AuNPs followed by L-cysteine to have a self-assembled monolayer. Subsequently, specific nanobodies for the surface epitope of *A. minutum* toxic strain were designed to form fusion proteins with SpyTag and covalently immobilized on the surface of the modified electrode via Spycatcher [81]. The SpyTag/SpyCatcher system allows the orientation of the nanobodies to rotate freely on the surface of the electrode that optimizes antigen-capture orientation for high target sensitivity [82]. Moreover, electrochemical impedance spectroscopy (EIS) technique was used to quantify *A. minutum* cells present in water samples by measuring the charge-transfer resistance of the samples using a hexacyanoferrate probe.

This particular immunosensor demonstrated a linear range of 10^3^–10^9^ cell L^−1^ with a detection limit of 3 × 10^3^ cell L^−1^, which exhibited a higher sensitivity when compared to other previously published diagnostic methods for label-free detection of *A. minutum* (Table 3). The higher sensitivity of this immunosensor is influenced by the functionalized nanobodies which are *nanomaterials*-based electrochemical immunosensor functionalized with nanobodies that can optimize the target selectivity due to the lower mass of nanobody (14 kDa) as compared to a conventional antibody (150 kDa for IgG). This will ultimately enhance the binding strength between *A. minutum* antigen and the nanobody [83]. Furthermore, the sensitivity of impedance signal variation detection will also increase due to the huge binder and antigen mass difference [72]. In addition, implementation of nanobodies in biosensors also offers other advantages compared to conventional antibodies, and this includes (i) cheap recombinant expression, (ii) small size, (iii) high solubility, (iv) high thermal and chemical stability, and (v) easy genetic manipulation for biotechnological application [84].

Based on EIS characterization, the fabrication of glassy carbon electrodes with AuNPs could create a minor resistivity of electron transfer [72] as they contain thousands of atoms that can be electrochemically reduced or oxidized and thereby act as a mediator to improve electron transfer [85]. The electrode was further fabricated with L-cysteine for the self-assembling layer and the carboxylated electrode surface was activated with EDC/NHS to significantly enhance the electron transfer resistivity. A further increment of electron transfer resistivity indicated the successful interaction of nanobodies with antigens. 

Additionally, another AuNPs-based immunosensor was developed to *A. minutum detection* [86]. This immunosensor consisted of AuNPs conjugated with captured antibody immobilized on a screen-printed carbon electrode (SPE) along with horseradish peroxidase conjugate as the detection reagent. A detection limit of 1 cell mL^−1^ with a 100% selectivity to detect *A. minutum* was achieved [86]. Furthermore, a direct sandwich ELISA method was carried out to detect *A. minutum* and the detection limit was roughly 5 cells mL^−1^. However, the 1 cell mL^−1^ detection limit was achieved when the sandwich format of an ELISA was replaced with AuNPs coated antibody on a sensor platform. The improved limit of detection can be explained due to the binding of antigen–antibody, which generates an electro-ionic signal that can be amplified with the incorporation of AuNPs in the sensor [87]. Hence, the presence of AuNPs can improve the detection limit as in the previously mentioned immunosensor. Additionally, AuNPs is known to be capable of conjugating with all kinds of biological molecules without altering their biochemical activity [88,89]. In terms of specificity, the immunosensor showed no cross-reactivity with algae that are non-toxic, however the sensor could detect *Alexandrium* species with more than 50% cross-reactivity [86]. When compared to biosensor designs in Table 3 and other assays (Table 4), the two nanomaterial-based immunosensors mentioned generally offer a faster detection time, wider linear range, and lower detection limits.

### 4.2. Nanomaterial-Based DNA Biosensor

Nanomaterials-based DNA biosensors have the potential to be the modern identification methods to detect HAB for all purposes. This type of biosensors will be more convenient, highly efficient, and a rapid yet accurate identification method for the detection of toxic algae species. The sensor can be fabricated from noble metal nanomaterials (e.g., Au, Al, Ag, Cr, Zn, CdS, Pt, Rh, Ir, Os, ZnO, SiO_2_, TiO_2_, etc.) and composites with different nanomaterial, latex, nanohybrids, conducting polymer matrix, or array immobilizing by the self-assembly of single-stranded probe DNA (ssDNA) on the latex or nanocomposite platforms [93,94,95]. The general procedures involved in the fabrication of electrochemical biosensors and their application in the detection of target DNA/protein, when nanoparticle labels are used for signal amplification. These sensors are capable of detecting extremely low levels of aquatic harmful algae species, which are associated with higher number of ssDNA molecules in the sample. There are mainly two purposes of using nanomaterials in DNA biosensors, i.e., as the substrate for DNA attachment and signal amplifier for DNA hybridization. The usefulness of this novel biosensing device will be examined by using it to analyse several organisms having an important impact on the global aquaculture industries [95]. 

A DNA optical biosensor fabricated with nanosheets made of graphene oxide (GO) was used to detect *Heterosigma akashiwo* (*H. akashiwo*) harmful algae [96]. The GO nanosheets assay was designed [97] and characterized with Fourier-Transform infrared (FTIR) spectrum [98]. This DNA sensing platform utilized GO nanosheets that have a strong interaction with fluorescent molecules, i.e., fluorescein amidite (FAM)-labelled probe, which resulted in the quenching of the fluorophore fluorescence. The fluorescence was quenched because of the energy transfer between the fluorescent dye and GO nanosheets was diminished. Once target DNA from the *H. akashiwo* hybridized with FAM-labelled probe and formed dsDNA, the fluorophore fluorescence was restored due to the desorption of dsDNA from the surface of GO nanosheets. The fluorescence intensity can be analysed at an emission wavelength of 480 nm and allowed the concentration of the harmful algae DNA fragments to be quantified and detected within 45 min. The detection time is an improvement when compared to other biosensors not fabricated with nanomaterial (Table 3). Furthermore, samples with 1 or 2 DNA mismatched bases could cause the fluorophore to fluoresce. However, the fluorescence intensity was significantly lower than the intensity produced by target DNA [96]. 

The GO nanosheets allows the transfer of efficient energy in a high manner that resulted in a detection limit of 1 pM specific genes of *H. akashiwo*, which was equivalent to 126.7 cells mL^−1^ [96]. In addition to the high energy transfer efficiency of graphene, it also has other advantageous properties, such as large surface area and biocompatibility [99]. Additionally, GO nanosheet sensor can also capture light for a longer period of time when compared to traditional sensors and is thereby suitable to be implemented in optical biosensors as well [100]. 

## 5. A Detailed Comparison of Biosensor and Conventional Methods for Harmful Microalgae Determination

A detailed comparison of various conventional methods for harmful microalgae determination with biosensor methods is summarized in Table 4. In general, conventional methods such as FISH PCR, ELISA and microarray demonstrated good specificity and sensitivity for the quantitative determination of harmful algae but they suffered the disadvantages of being expensive and not portable as they could only operate in a laboratory environment. Due to the sophistication of these methods when compared to biosensors, they tend to take a longer time to obtain rapid test results. Biosensors on the other hand are not only portable but also can yield rapid test results without compromising the accuracy of the determination. These biosensor devices for microalgae determination are relatively new and production technology is not matured, thus the reproducibility performance of the devices may be an issue. Considering the benefits they provided in comparison with conventional methods, it is envisaged that biosensor technology can be a good alternative for HAB monitoring and management in the near future. 

## 6. Conclusions

Harmful algal bloom events have been increasingly reported all over the world and many conventional methods have been used for the management of these types of environmental issues. Some progress in the conventional methods, such as the microarray approach had led to the detection of thousands of samples in a single test and this could allow reliable simultaneous detection of HAB. However, there is still potential for rapid identification methods that promise fast or easier handling in terms of the detection and monitoring of HAB worldwide. New techniques based on biosensors for the detection of HAB are an improvement over some conventional detection methods, especially with the implementation of nanomaterials in electrochemical biosensors, which can improve the simplicity of detection, sensitivity, and detection time have led to a reliable fast and simpler identification of harmful microalgae. Although optical-based biosensors are an area that is promising for rapid microalgae analysis, this area is less well explored for such an application. More studies should focus on designing and optimizing biosensors technology that could further improve biosensor reliability and reproducibility for HABs detection. Thus, better prevention, management, and mitigation strategies can be adopted by the stakeholders and relevant authorities to minimize the negative impacts of HABs. 

## Figures and Tables

**Figure 1 sensors-22-03144-f001:**
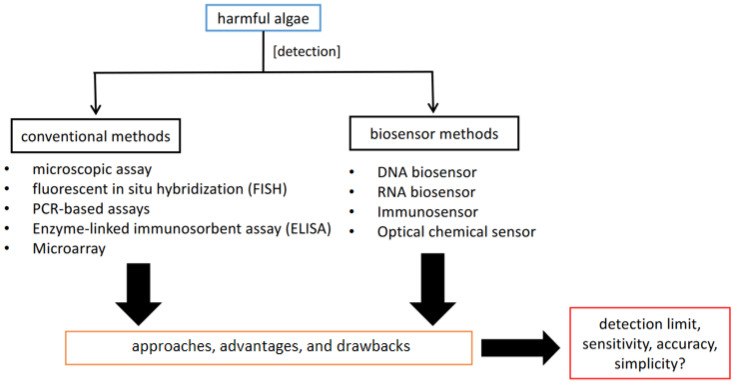
General overview of the review paper.

**Figure 2 sensors-22-03144-f002:**
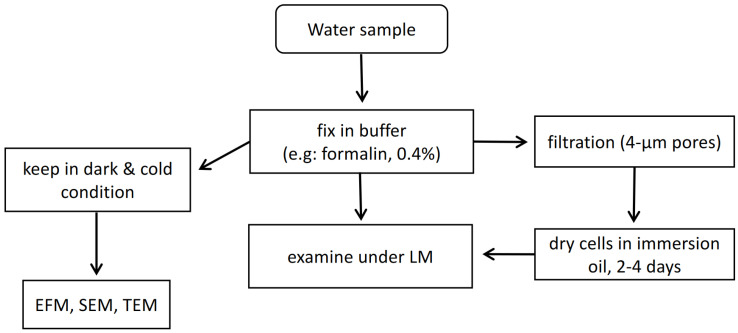
Simplified diagram of microscopy methods in HAB examination.

**Figure 3 sensors-22-03144-f003:**
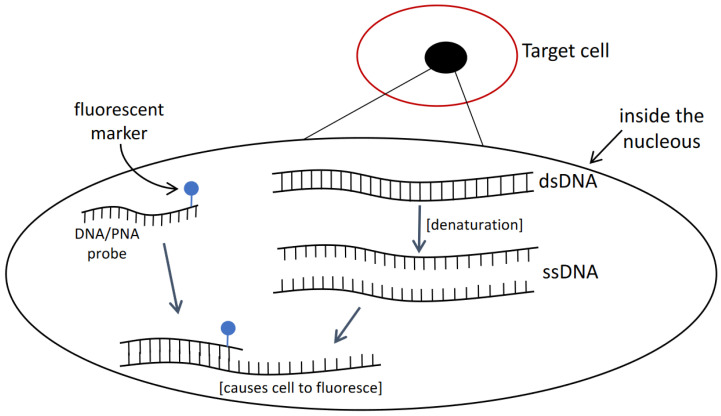
Principle of fluorescence in situ hybridization (FISH) technique. A method that utilizes fluorescent labelled probes to cause target cells to fluoresce.

**Figure 4 sensors-22-03144-f004:**
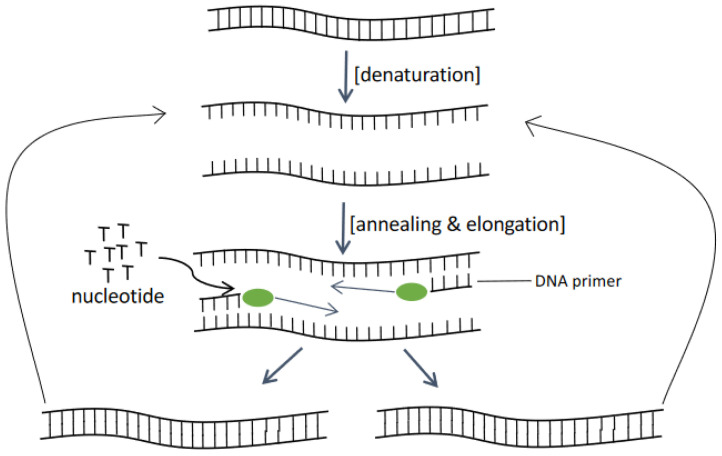
General concept of PCR.

**Figure 5 sensors-22-03144-f005:**
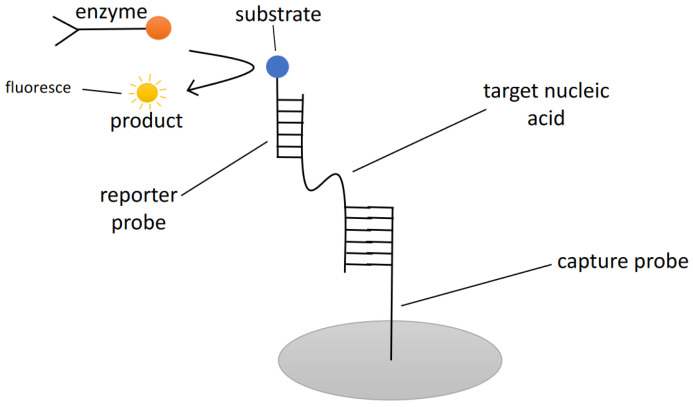
An example of enzyme-linked immunosorbent assay (ELISA) concept. A sandwich hybridization concept with a measurable output based on fluorescence.

**Figure 6 sensors-22-03144-f006:**
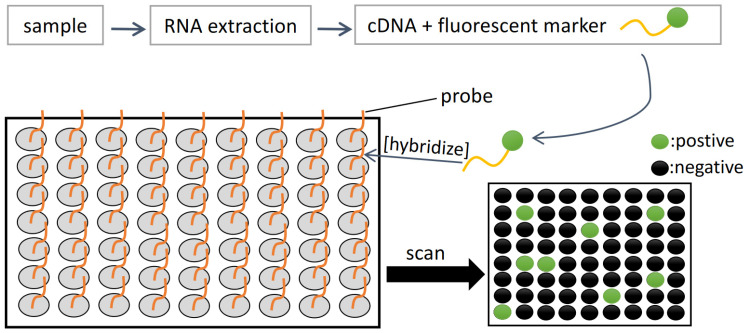
Basic principle of microarray.

**Figure 7 sensors-22-03144-f007:**
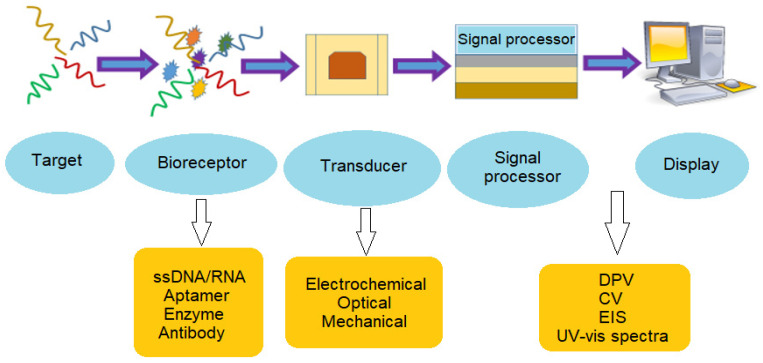
Schematic diagram of biosensor functionality.

**Figure 8 sensors-22-03144-f008:**
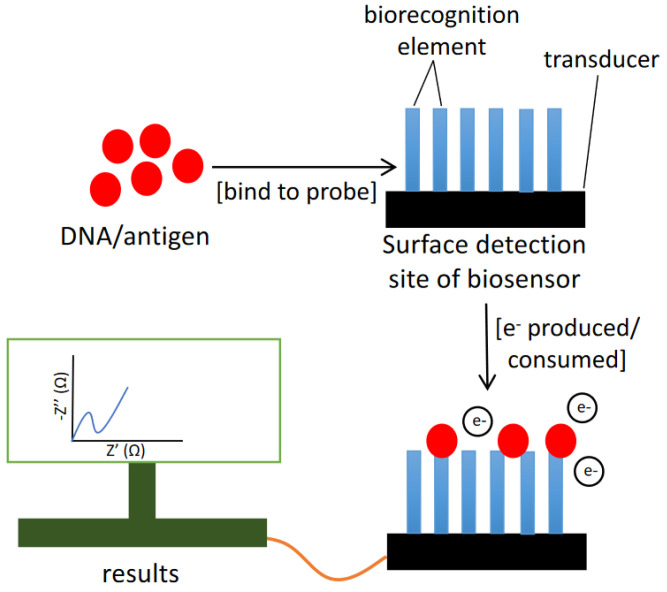
Basic concept of electrochemical biosensor.

**Figure 9 sensors-22-03144-f009:**
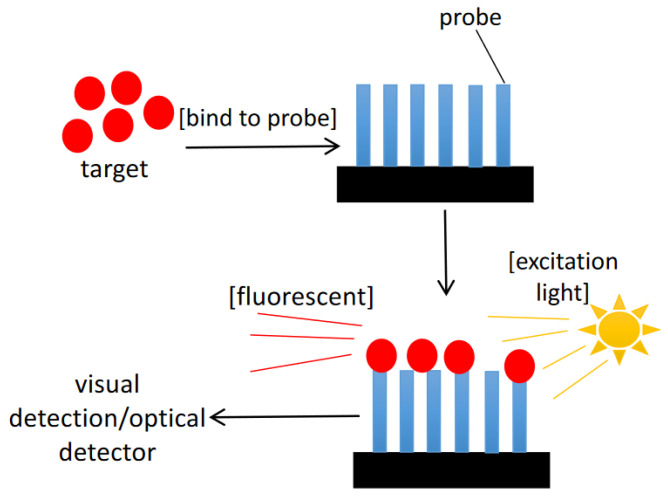
General principle of optical sensor.

**Table 1 sensors-22-03144-t001:** Summary of the previously reported molecular biological methods for the detection of HAB species.

Reference	Target	Instruments/Methods	Response Time	Detection Limit	Advantages	Drawbacks
[11,36]	Autotrophs	Light Microscopy	2 h–4 days	-	Useful for the studies of taxanomy and morphology of whole HAB cells.	Incapable of distinguishing cells below 5 μm in size. Possible human errors in counting cells.
[36]	Autotrophs	TEM and SEM	1–10 days	-	Allows identification of pico- and nano-sized organisms.	Time-consuming, expensive.
[44]	*Prorocentrum donghaiense*	FISH	1 h	-	An accurate detection method. Usage of PNA probe displayed more intensive green fluorescence than DNA probe.	The quantification requires LM and is thereby slow to produce results and possibly comes with human error in counting.
[43]	*Alexandrium minutum*	FISH	45 min	-	A rapid detection tool for *A. minutum*	Quantification requires LM. Auto-fluorescence leads to false positive results. Challenge in fixing microalgae cells.
[45]	*Karlodinium veneficum* (Kv), *Chattonella marina* (Cm), *Skeletonema* spp., *Scrippsiella trochoidea* (St), *Karenia mikimotoi* (Km), and *Prorocentrum donghaiense* (Pd)	Multiplex PCR (mPCR)	-	6 cells per reaction (*Skeletonema* spp., Cm, and St) 60 cells per reaction (Km, Pd, and Kv)	Allows simultaneous multiple targets detection. Strong specificity and stability.	Numerous optimization steps are required to have a good mPCR performance. Lack of accuracy in determining cell density.
[49]	*Alexandrium* spp.	qPCR	-	<5 × 10^3^ cells L^−1^	High sensitivity of detection.	Overestimation of abundance.
[23]	*A. minutum*	ELISA	<3 h	1 × 10^4^ cells sample^−1^	Capable of detecting *A. minutum* cells at different cell counts in the presence of a complex background.	Require at least 10,000 cells for measurable RNA concentration, based on the extraction kit used in this experiment. Enumeration of cell count is based on estimation only.
[50]	*A. minutum* species	Whole-cell ELISA	-	1 × 10^5^ cells L^−1^	Good sensitivity and specificity on natural seawater samples.	Generation of monoclonal antibodies via rats. Tends to overestimate the number of cells by a rough factor of 10.
[51]	*A. fundyense,* *Pseudo-nitzschia australis,* and *A. Ostenfeldii*	Fiber optic microarrays	-	5–10 cells sample^−1^ (*A. fundyense* and *Pseudo-nitzschia Australis)* 50 cells sample^−1^ (*A. Ostenfeldii)*	Simultaneous detection of all three species. Simple and reusable sensor with no loss of sensitivity.	Complex instrumentations with microscopic epifluorescence and image analysis. High experimental and set-up costs.
[52]	*A. fundyense,* *Pseudo-nitzschia australis,* and *A. Ostenfeldii*	Fiber optic microarrays	45 min	5 cells sample^−1^	Simultaneous detection with no cross-reactivity.	High experimental and set-up cost.

**Table 2 sensors-22-03144-t002:** Summary of the previously reported biosensors methods for the detection of HAB species.

Reference	Target	Instruments/Methods	Response Time (h)	Detection Limit (Cells L^−1^)	Advantages	Drawbacks
[62]	DNA/RNA of microbial pathogens	Rapid PCR-Detect and Hybrid PCR-Detect.	4–6	-	Sensitive detection of sample DNA/RNA.	Only capable of detecting single-base mutations from pure culture isolate.
[64]	rRNA of toxic algae (toxic dinoflagellate *A. Ostenfeldii*)	Molecular DNA probes	7–10	5 × 10^9^	Simplified detection methods	Manual RNA isolation and manipulation of the hybridization steps are required at high temperature system.
[63]	Microbial pathogens and *Karenia brevis*	8-plex assay of microbes	3–5	1000	Able to multi-target electrochemical detection of microbial pathogens.	Complex steps in DNA extraction.
[23]	rRNA of harmful algae species	Multi-probe chip and a semi-automated rRNA biosensor	~2	-	Allows in situ detection and monitoring of toxic algae.	Manual rRNA isolation.
[35]	*Alexandrium minutum*	Multi-probe biosensor (ALGADEC)	~2	25,000	Almost fully automated device for in situ analysis.	Poor limit detection.
[65]	*Prymnesium parvum, Gymnodinium catenatum, and Pseudo-nitzschia australis*	DIG-enzymatic label assay	1–2	-	Simple and easy handling amperometric techniques.	Very poor response for cyclic voltammetry.
[66]	*Alexandrium* species	Surface Plasmon Resonance (SPR) biosensing instrument and peptide nucleic acid probes	>3.5	-	Cost effective and yield quick result.	Require tubing flushing maintenance.

**Table 3 sensors-22-03144-t003:** Comparison of several *A. minutum* detection methods by biosensors/assays.

Reference	Detection Method	Detection Time (min)	Linear Range (Cells L^−1^)	Detection Limit (Cells L^−1^)
[72]	Electrochemical nanobody immunosensor	≤75	5.00 × 10^3^–1.00 × 10^9^	3.1 × 10^3^
[86]	AuNPs-based Immunosensor	≤30	-	1 × 10^3^
[90]	Loop-mediated isothermal amplification assay	≤120	≈1.00 × 10^4^–1.00 × 10^8^	≈1.7 × 10^4^
[91]	Quartz crystal microbalance	≤80	1.50 × 10^9^–5.50 × 10^9^	1 × 10^9^
[92]	Super-paramagnetic immunochromatographic strip test	≤30	≈2.00 × 10^5^–2.45 × 10^8^	5 × 10^4^

**Table 4 sensors-22-03144-t004:** General comparison of biosensor and conventional methods in HAB detection.

Methods	Advantages	Drawbacks	* Detection Time	Portablity	Ease of Operation
Microscopic assay	Useful for taxonomy and morphology study	Time consuming and expensive	Very slow	Lab based	Fairly easy
FISH	Accurate detection method	Quantification requires a microscopy approach	Moderate	Lab based	Fairly easy
PCR	High specificity of detection	Numerous optimization steps are required for good test results	Slow/Moderate	Lab based	Complicated
ELISA	Good specificity even in the presence of complex background	Expensive	Slow	Lab based	Complicated
Microarray	Allows numerous simultaneous detections, good sensitivity	Expensive	Moderate	Lab based	Complicated
Electrochemical Biosensor	Small, simple, robust devices, and good detection limits	Device less reproducible	Fast	On-site	Easy
Optical Biosensor	High specificity and cost-effective	Device less reproducible	Fast	On-site	Easy

* Detection time is based on word descriptions, this includes very slow (several hours to days), slow (few hours), slow/moderate (few hours to several hours), moderate (less than few hours), and fast (less than an hour).

## Data Availability

Not applicable.

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
