# Peer review of "Harmful Microalgae Detection: Biosensors versus Some Conventional Methods"

_sensors, 2022, doi:10.3390/s22093144_

Round 1

Reviewer 1 Report

This review presents an overview of the state-of-art of development of biosensor technology for harmful microalgae detection, compared with conventional methods especially those involved in alga bloom environment (HAB). The topic of the manuscript is quite inflated, in fact, in the literature are presents several review papers about this subject. The report gives an updated and almost detailed picture on this research, describing the several approaches followed in the several papers taken into account. In my opinion, a more in-depth comparison of the main features and drawbacks of methods could be useful to have a complete picture of the topic taken into account. For this reason, before proceeding to a possible acceptance of the manuscript for publishing in this journal, a minor revision of the paper is necessary.

Author Response

  1. This review presents an overview of the state-of-art of development of biosensor technology for harmful microalgae detection, compared with conventional methods especially those involved in alga bloom environment (HAB). The topic of the manuscript is quite inflated, in fact, in the literature are presents several review papers about this subject. The report gives an updated and almost detailed picture on this research, describing the several approaches followed in the several papers taken into account. In my opinion, a more in-depth comparison of the main features and drawbacks of methods could be useful to have a complete picture of the topic taken into account. For this reason, before proceeding to a possible acceptance of the manuscript for publishing in this journal, a minor revision of the paper is necessary.

ANSWER- Response to Reviewer #1

Chapter 5 (A detailed comparison of biosensor and conventional methods for harmful microalgae determination) had been created to complete the picture of this topic. The following information and Table 4 can be found on page 20, line 529-548 of the manuscript file:

A detailed comparison of various conventional methods for harmful microalgae determination with biosensor methods is summarized in Table 4. In general, conventional methods such as FISH PCR, ELISA and microarray demonstrated good specificity and sensitivity for the quantitative determination of harmful algae but they suffered the disadvantages of being expensive and not portable as they could only operate in a laboratory environment. Due to the sophistication of these methods when compared to biosensors, they tend to take a longer time to obtain rapid test results. Biosensors on the other hand are not only portable but also can yield rapid test results without compromising the accuracy of the determination. These biosensor devices for microlagae determination are relatively new and production technology is not matured, thus the reproducibility performance of the devices may be an issue. Considering the benefits they provided in comparison with conventional methods, it is envisaged that biosensor technology can be a good alternative for HAB monitoring and management in the near future.

Reviewer 2 Report

Manuscript ID:  sensors-1661301-peer-review-v1

Title: Harmful Microalgae Detection: Biosensors Versus some Conventional Methods

This paper is a review that summarizes comparatively different methodologies for Harmful microalgae detection. It goes from Conventional to Biosensor-based methods, including those that use nanoparticles to improve selectivity and increase sensitivity. In my opinion, it could be published after some revisions. Authors has not discussed results obtained using optical biosensors, and biosensors based on the use of Nanoparticles with optical detection.

With the aim of completing this review I would suggest:

  • To include in Section 3.2 Optical biosensors. Some analytical characteristics of these type of sensors (limit of detection, selectivity, complexity or not of sample pre-treatment, etc.).
  • 1 Nanomaterial-based immunosensors. All the data included in this item are about electrochemical sensors, please, include at least a paragraph with alternatives of analysis using optical biosensors.

Page 7 line 241: U1PLC-MS/MS. What number 1 means in this abbreviation?

Author Response

Title: Harmful Microalgae Detection: Biosensors Versus some Conventional Methods. This paper is a review that summarizes comparatively different methodologies for Harmful microalgae detection. It goes from Conventional to Biosensor-based methods, including those that use nanoparticles to improve selectivity and increase sensitivity. In my opinion, it could be published after some revisions. Authors has not discussed results obtained using optical biosensors, and biosensors based on the use of Nanoparticles with optical detection. With the aim of completing this review I would suggest:

  1. To include in Section 3.2 Optical biosensors. Some analytical characteristics of these type of sensors (limit of detection, selectivity, complexity or not of sample pre-treatment, etc.).

ANSWER - The majority of the published optical biosensors in HAB detection are more towards detecting the HAB’s toxin content, rather than the harmful algae species itself.  However we have included here in this review optical biosensor technology and one reported example of its usage for microalgae species analysis (please see page 16, line 406-423):

3.2. Optical biosensors

Optical chemical sensors have also been employed to detect HAB [78]. Optical chemical sensors are based on the reaction between biomolecule compounds and analytes, where the optical characteristics are induced by UV-vis absorption, bioluminescence, chemiluminescence, fluorescence, or reflectance. These biosensors have the benefits of being flexible, compact, and resistant to electrical noise (Figure 9) [79]. Furthermore, thanks to the use of optical fibre, it is feasible to construct very compact sensors, making them suited for measurement [80, 81].

Unlike toxin from microalgae, to date there is not many optical biosensors for the determination of microalgae species being reported. So far, Alexandrium species had be identified via an optical technique known as SPIRIT+, which utilizes portable surface plasmon resonance (SPR) biosensing equipment and peptide nucleic acid (PNA) probes [82]. SPR is a label-free technology that measures changes in the refractive index when a target sequence is hybridized on a surface. The resonance unit (RU) was used to quantify changes in the refractive index, where one RU is equal to 1×10-6 degrees. Additionally, SPIRIT is claimed to be a cost-effective and user-friendly approach that yields quick results.

  1. 1 Nanomaterial-based immunosensors. All the data included in this item are about electrochemical sensors, please, include at least a paragraph with alternatives of analysis using optical biosensors.

ANSWER- So far, we have not found immunosensors based on optical biosensor for the determination of microalgae species. Limited journals can be found with the use of optical biosensors incorporated with nanomaterial but mostly for the detection harmful algae’s toxin content, which is not the focus of this review. However, we provided an example of optical biosensor for harmful microalgae determination using SPR concept as below (Please see page 16, line 413-420):

Unlike toxin from microalgae, to date there is not many optical biosensors for the determination of microalgae species being reported. So far, Alexandrium species had be identified via an optical technique known as SPIRIT+, which utilizes portable surface plasmon resonance (SPR) biosensing equipment and peptide nucleic acid (PNA) probes [82]. SPR is a label-free technology that measures changes in the refractive index when a target sequence is hybridized on a surface. The resonance unit (RU) was used to quantify changes in the refractive index, where one RU is equal to 1×10-6 degrees. Additionally, SPIRIT is claimed to be a cost-effective and user-friendly approach that yields quick results.

  1. Page 7 line 241: U1PLC-MS/MS. What number 1 means in this abbreviation?

ANSWER-The word “U1PLC-MS/MS” in page 7 line 241 is an unintentional mistake. The correction has been made to change the word “U1PLC-MS/MS” to “UPLC-MS/MS”.

This manuscript is a resubmission of an earlier submission. The following is a list of the peer review reports and author responses from that submission.